# Subgiants in NGC 188 reveal that rotationally induced mixing creates the main sequence Li-Dip

Qinghui Sun [1,2] ✉, Constantine P. Deliyannis [3], Barbara J. Anthony-Twarog [4], Bruce A. Twarog [4], Xiao-Tian Xu [5,6], Aaron Steinhauer [7] & Jeremy R. King [8]

The Lithium-Dip is a severe lithium depletion observed in mid-F (6200-6650 K) dwarfs, which has puzzled astronomers since it was discovered in 1986. Proposed mechanisms include effects related to rotation, magnetic fields, diffusion, gravity waves, and mass loss. Which, if any, of these is realistic remains unclear. Here we show that mixing due to shear induced by stellar angular momentum loss is the unique mechanism driving the lithium depletion. Each mechanism leaves a different signature in the subsurface lithium distribution. The deepening surface convection zones of subgiants of NGC 188 evolving out of the Lithium-Dip dredge up the subsurface material and thus reveal the signature of the responsible mechanism, rotation. Subgiants can also be used more generally, thereby improving fundamental understanding of stellar evolution. Rotational mixing may be the dominant lithium-depleting mechanism in a wide range of solar-type stars, including in the Sun. Our results may further reconcile the cosmological lithium discrepancy.

Inside main-sequence stars (dwarfs), Lithium (Li), Beryllium (Be), and Boron (B) survive only in the outermost, coolest layers, to progressively greater depths. Their observed surface abundances thus provide invaluable information about physical mechanisms occurring below. The "standard" theory of stellar evolution (SSET[1–3]) ignores complicated but potentially interesting effects due to rotation, magnetic fields, diffusion, and mass loss. The SSET predicts that G dwarfs like the Sun and even more so early K dwarfs will have depleted some Li due to convective mixing to deep, hot layers where Li is destroyed by energetic protons—and will have done so only during the pre-main sequence. More massive stars like F and A dwarfs will have suffered a negligible Li depletion. In sharp contradistinction to the SSET, a severe Li depletion was discovered in mid-F dwarfs (ref. [4], the "Li-Dip," Fig. 1). This Li-Dip is an increasingly steep depletion of Li with increasing $T_{eff}$, from relatively high levels of A(Li) at 6200 K to unobservably low levels of A(Li) at 6650 K, followed by a very sudden and very sharp increase of

A(Li) from 6650 to 6750 K. We label this nearly vertical feature "the Wall." The Li-Dip provided the first fully convincing evidence that the SSET was missing important physics. Attempts to identify the missing physics quickly proliferated, and included mechanisms such as diffusion[5,6], mass loss[7], and slow mixing due to effects either of gravity waves[8] or rotation[9,10]. Between approximately 1986 and 2010, it became clear that G and K dwarfs also deplete Li during the main sequence[11,12], which is highly non-standard, and in 2019, it was discovered that A dwarfs also do so[13]. Although the standard model of the current Sun, a G2V dwarf, agrees very well with helioseismology (e.g.,[14,15]), the predicted Li depletion of only a factor of three ("X" in Fig. 1) underestimates the actual depletion by a factor of over 50. This enormous discrepancy highlights a major limitation of the SSET, and we need to understand the secret life that the Sun has led. In fact, nearly all dwarfs for which it is possible to observe the surface Li abundance deplete Li over time in a non-standard way. Understanding

[1]Tsung-Dao Lee Institute, Shanghai Jiao Tong University, Shanghai, China. [2]Department of Astronomy, Tsinghua University, Beijing, China. [3]Department of Astronomy, Indiana University, Bloomington, IN, USA. [4]Department of Physics and Astronomy, University of Kansas, Lawrence, KS, USA. [5]School of Astronomy and Space Science, Nanjing University, Nanjing, Jiangsu, China. [6]Argelander-Institut für Astronomie, Universität Bonn, Bonn, Germany. [7]Department of Physics and Astronomy, State University of New York, Geneseo, NY, USA. [8]Department of Physics and Astronomy, Clemson University, Clemson, SC, USA. ✉e-mail: qinghuisun@sjtu.edu.cn

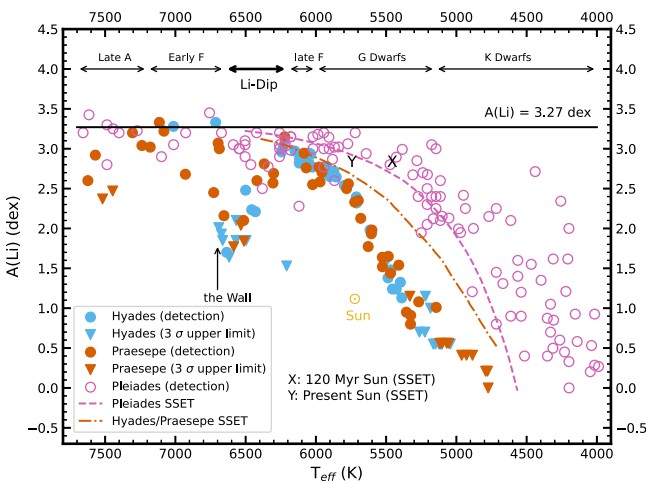

**Fig. 1 | Each cluster in this figure provides a snapshot of how, at a given age, A(Li) depends on mass, or equivalently $T_{eff}$ for main-sequence stars.** Only single members are shown, as binaries can lead to various errors and misinterpretations[45,66]. Detections are shown as circles, while $3\sigma$ upper limits are shown by downward arrows. Pleiades, Hyades, and Praesepe stars are color-coded in magenta, blue, and orange-red, respectively. Regions corresponding to Late A to K dwarfs are labeled, with the Li-Dip marking pronounced depletion in early F stars. The SSET predictions are approximately consistent with the lower envelope in the Pleiades (age about 120 Myr). However, the SSET fails to explain the observed scatter, especially in cooler Pleiads[3,67], and notably fails to explain the main sequence Li depletion exemplified by Hyades and Praesepe (ages about 650 Myr). Also shown are the meteoritic abundance at A(Li) = 3.27 dex[68], which is the presumed initial Solar abundance, the value predicted from the SSET for the current Sun (age about 4.5 Gyr[69], Y; X indicates this A(Li) at the $T_{eff}$ the Sun would have had at 120 Myr using Yonsei-Yale ($Y^2$) isochrones[46], to compare to the Pleiades), and the actual current Solar A(Li)[70], which lies fully a factor of 150 below the meteoritic abundance. Pleiades data are from refs. [71], [72], as placed on consistent $T_{eff}$ and A(Li) scales by ref. [45], while Hyades and Praesepe data are from ref. [3] and include their own data plus those compiled therein from refs. [4,66,73–75], which were placed by them on the same consistent $T_{eff}$ and A(Li) scales. We show the SSET model predictions of A(Li) for slow rotators in the Pleiades[22] and in the Hyades and Praesepe clusters[3]. While the SSET model reproduces the A(Li) trends of Pleiades slow rotators well, it fails to capture key physical processes in older clusters like Hyades and Praesepe, leading to discrepancies with observations. Source data are provided as a Source data file.

the physical cause of these Li depletions is thus of fundamental importance to improving our knowledge of stellar evolution. G and K dwarfs and even A dwarfs also spin down during the main sequence, which suggests a possible connection between stellar spindown and Li depletion[13]. For the Li-Dip, Be and B data discriminate between scenarios because whereas all the proposed mechanisms create a Li-Dip, the ratios between Li, Be, and B differ from mechanism to mechanism. The correlations between Li and Be depletion in dwarfs[16,17] and between Be and B depletion[18–20] have argued strongly against diffusion, mass loss, and slow mixing due to gravity waves, and favor a specific kind of rotationally induced mixing, namely, that arising in models where angular momentum loss, which results in stellar spindown, triggers an internal shear instability that results in internal angular momentum transport and mixing ("Yale models"[9,21,22]). In this paper, we show that it is possible to use Li alone, in a very different and highly complementary way to discriminate between scenarios, by studying Li in subgiants evolving out of the Li-Dip.

The Li preservation region is a key discriminator because each of the proposed mechanisms creates a different Li profile (dependence of the Li abundance with depth[23]). It is instructive to use the SSET as a reference: In the SSET, nearly the entire Li preservation region is radiative, so no mixing occurs except inside the surface convection zone (SCZ). The SCZ is much shallower than the Li preservation region,

so convection plays almost no role in modifying the Li profile. The Li abundance remains constant as a function of depth until the temperature is high enough to destroy Li; then the Li abundance drops very sharply with depth (the "Li preservation boundary"). In the mass loss scenario, nearly all of the Li preservation region is lost, so that the Li abundance already drops very sharply with depth immediately below the very shallow SCZ. In the diffusion scenario, the downward diffusion time scales increase with depth, which causes Li to increasingly pile up with depth until the Li preservation boundary is reached. In the Yale models, slow rotational mixing reduces the amount of Li in the Li preservation region compared to the SSET, and, importantly, since the efficiency of the shear-induced mixing decreases with depth, the Li profile is steeper than in the SSET but shallower than in the mass loss scenario.

Subgiants evolving out of the Li-Dip can reveal which of these distinct Li profiles is most realistic[23], if any. This is because, as subgiants evolve to lower surface temperatures ($T_{eff}$), their SCZs deepen considerably, eventually dredging up the entire Li preservation region. Thus, the Li-$T_{eff}$ relation in star clusters with well-populated subgiant branches determines the true shape of the Li profile. Note that although subgiants evolve to a large range of $T_{eff}$, the entire subgiant branch in a given cluster has evolved out of a very small range of $T_{eff}$ (mass) of the main sequence. In the SSET, the surface A(Li) stays constant from the main sequence turnoff to lower $T_{eff}$ until the SCZ reaches the Li preservation boundary. As the SCZ deepens further with lower $T_{eff}$ and mixes the now fixed amount of Li with regions containing no Li, A(Li) decreases through "subgiant dilution" until the SCZ contains most of the mass of the star, on the red giant branch. The total amount of dilution is nearly 2 dex[24]. In the mass loss scenario, A(Li) declines dramatically with lower $T_{eff}$, starting immediately with evolution past the turnoff. In the diffusion scenario, A(Li) initially increases with lower $T_{eff}$ as the SCZ dredges up increasingly piled up Li, until subgiant dilution. For the Yale rotational scenario, the Li-$T_{eff}$ trend has some negative slope but not as steep as in the mass loss scenario.

Several past studies have favored rotational mixing as the mechanism shaping Li depletion (e.g., [3,25]), with mass loss and diffusion shown to be less consistent with observations (e.g., [26]). Li data from star cluster M67 were possibly consistent with Yale rotational mixing, and provided strong evidence against diffusion, but could not rule out mass loss if, for example, the model stellar $T_{eff}$ scale was too high by order of 200 K[23]. This ambiguity was removed when the Be data in M67 were combined with the Li data, ruling out mass loss and strongly supporting rotational mixing[27]. However, Be data can be very challenging to obtain, requiring much observing time on only a few of the world's largest telescopes that are suitably equipped with a UV-capable high-resolution spectrograph. For example, the M67 Be data required data acquisition at the Keck I 10-m telescope over a period of 4 years, for only 9 stars; this is the most efficient telescope in the world for stellar Be work. B data are even more challenging to obtain, requiring the use of the Hubble Space Telescope. Moreover, Be observations are limited to far brighter stars than can be observed effectively for Li, and B observations are limited to even brighter stars. For example, M67 subgiants are much too faint for B observations. While multi-element studies have managed to provide some evidence supporting rotational mixing as the primary depletion mechanism to create the Li-Dip, it remains of great interest to develop methods where Li data alone, without additional information from Be and/or B data, suffice to give us a reasonably full picture.

Here, we show that Li observations of subgiants in NGC 188 support rotational mixing as the primary cause of the Li-Dip, using Li data alone, and definitely rule out other mechanisms. These data extend Li-Dip studies to an older cluster than M67, expanding the observational sample for Li-Dip studies using subgiants, while also opening up great opportunities for future Li-Dip studies toward even older clusters.

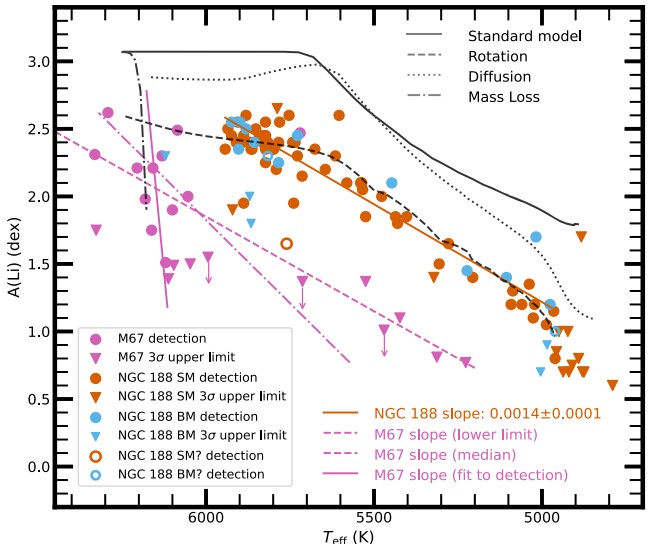

**Fig. 2 | A(Li) versus $T_{eff}$ for turnoff stars and subgiants in M67 and NGC 188.** The single members (SM), binary members (BM), single likely members (SM?, where "?" denotes uncertain membership status), and binary likely members (BM?) of NGC 188 are shown in colored symbols. Members of M67 from ref. 23, with membership probability >0.5 from ref. 65, are also shown. Detections are shown as circles, and 3σ upper limits as downward triangles. The best linear fit to NGC 188 detections is shown as an orange-red line. The standard model for NGC 188, along with models that include rotational mixing, diffusion, and mass loss, is shown with gray lines and denoted in the figure, based on MESA simulations described in detail in the Methods section. For M67, a solid magenta line shows the linear fit to detections only, while a dashed line adopts the best-fit slope from NGC 188 as a lower limit. The magenta dot-dashed line shows the median slope from a probabilistic model fit that includes the M67 upper limits. Downward arrows further show that the true A(Li) values lie below the plotted upper limits. Source data are provided as a Source data file.

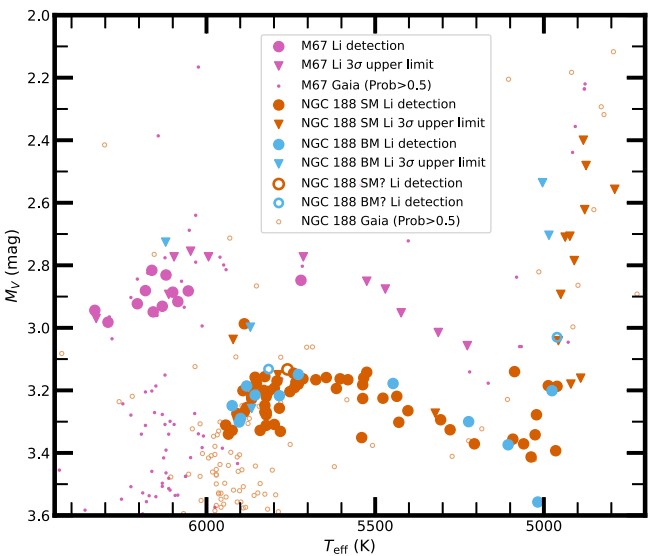

**Fig. 3 | Hertzsprung–Russell diagram of stars from M67 and NGC 188.** Small dots indicate members of M67 (magenta) and NGC 188 (orange-red), selected based on a membership probability greater than 0.5 from ref. 65. The $V$ and $B−V$ values for Gaia members in M67 are retrieved by cross-matching with data from ref. 63, and for NGC 188 are by cross-matching with data from ref. 44. Larger symbols indicate stars selected in this study. Circles show stars with Li detections, whereas downward triangles show stars with Li upper limits. Single members (SM) and single likely members (SM?) of NGC 188 are in orange-red, whereas binary members (BM) and binary likely members (BM?) are in blue. Members of M67 from ref. 23, with membership probability >0.5 from ref. 65, are shown in magenta. Source data are provided as a Source data file.

## Results and discussion

NGC 188 is slightly older (age about 6 Gyr[28]) than M67 (age about 4.5 Gyr[29]), so its subgiants evolve out of a slightly cooler portion of the Li-Dip, and are slightly less massive than those in M67. To create a Li-Dip, the Li profile in the mass loss scenario remains very steep as a function of depth, and in the diffusion scenario, the Li abundances again increase at first with depth until the Li preservation boundary is reached. For rotational mixing, the Li profile is shallower than in M67 because of the smaller stellar mass[21]. The implications for subgiants in NGC 188 evolving to lower $T_{eff}$ are that (a) for mass loss, the Li abundances will again decline very steeply, (b) for diffusion, the Li abundances will again increase until dilution begins, and (c) for rotational mixing, the Li-$T_{eff}$ relation will be shallower than that in M67.

Figure 2 shows Li abundances for subgiants from the turnoff to undetectably low levels for M67[23] and for NGC 188 (presented in this study). The Li-$T_{eff}$ relation is shallower for NGC 188 than for M67. We show Modules for Experiments in Stellar Astrophysics (MESA) model curves for the standard, rotation, diffusion, and mass loss cases, using the metallicity (+0.064 dex) and turnoff mass (1.2 $M_{\odot}$) of NGC 188[28]. In comparison to model predictions, the shallow slope observed in NGC 188 does not align with mass loss. As stars evolve onto the subgiant branch, the observed A(Li) decreases with decreasing $T_{eff}$, whereas the standard and diffusion models suggest either a nearly flat trend or a slight increase in this region. The diffusion model also produces a slightly steeper slope than observations. Although the standard model shows a roughly similar slope to the observations at $T_{eff}$ < 5700 K, it does not reproduce the sharp decline in A(Li) around 5000 K, where A(Li) declines from detections to only upper limits within a small $T_{eff}$. In contrast, the rotation model better captures this drop near 5000 K. Additionally, both the standard and diffusion models retain too much Li at the start of the subgiant phase and predict minimal early depletion, contrary to the lower A(Li) values and immediate decrease in A(Li) seen in NGC 188 subgiants. All models adopt the same initial A(Li) of 3.25 dex, based on the meteoritic value[30], which suggests that the standard and diffusion models underestimate Li depletion during the pre-MS and MS phases. The rotational mixing model closely reproduces both the slope and the overall depletion pattern, offering the best agreement with the data. Moreover, the A(Li)−$T_{eff}$ trend in NGC 188 is shallower than in M67, further supporting a rotational mixing scenario. Thus, the Li-$T_{eff}$ relation for NGC 188 subgiants supports rotational mixing and argues against mass loss, diffusion, and the standard model.

Originally, it was Li data alone that signaled the striking effects of highly non-standard physical mechanisms acting in the outer layers of stars during the main sequence to deplete their surface Li abundance, creating, for example, the Li-Dip. Since then, the strongest evidence identifying rotationally induced mixing as the dominant responsible mechanism instead of other proposed ones, such as diffusion or mass loss, has come from the correlated depletion of Li with depletion of Be and B, and with stellar spindown. Finally, and importantly, as of 2025, nearly 40 years after the discovery of the Li-Dip, we have come full circle: these Li data alone point to rotationally induced mixing as the unique, dominant Li depletion mechanism creating the Li-Dip.

There is an interesting and potentially important connection to cosmology. The $T_{eff}$ range (6200−6000 K) of the NGC 188 subgiant progenitors coincides, in part, with the Li-$T_{eff}$ Plateau observed in metal-poor dwarfs[31,32]. These metal-poor stars formed with the Big Bang Li abundance, and a subsequent depletion by a factor of 3 would support the standard model of Big Bang Nucleosynthesis, but a different degree of depletion would point to a potentially serious problem in Big Bang cosmology[33,34]. In NGC 188, rotational mixing depleted the Li of these progenitors by about a factor of 10, as suggested by current turnoff stars (Figs. 2 and 3) and a presumed initial A(Li) near the meteoritic A(Li), or slightly higher to account for Galactic Li production[33,34] and NGC 188's super-solar metallicity. It is then

**Table 1 | Key stellar parameters for M67 members used in this study, including star ID, equatorial coordinates (RA, DEC; J2000), apparent V-band magnitude, B−V color, effective temperature (T_eff), and lithium abundance A(Li)**

| ID[a] | RA[a] (J2000) | DEC[a] (J2000) | V[a] | B−V[a] | $T_{eff}^{b}$ | A(Li)[b] |
|---|---|---|---|---|---|---|
| | h m s | ° ′ ″ | mag | mag | K | dex |
| 5123 | 132.67453 | 11.61483 | 12.647 | 0.685 | 5713 | <1.37 |
| 5041 | 132.64307 | 11.66580 | 12.629 | 0.594 | 6047 | <1.50 |
| 5042 | 132.64119 | 11.77392 | 12.856 | 0.531 | 6292 | 2.62 |
| 5118 | 132.66988 | 11.79610 | 12.818 | 0.521 | 6331 | 2.31 |
| 5061 | 132.65271 | 11.81544 | 13.025 | 0.574 | 6123 | 2.26 |
| 5191 | 132.69851 | 11.74733 | 12.700 | 0.482 | 6490 | <2.22 |
| 5228 | 132.70811 | 11.82027 | 12.931 | 0.851 | 5227 | <0.77 |
| 5362 | 132.75447 | 11.83643 | 12.725 | 0.739 | 5525 | <1.37 |
| 5284 | 132.72659 | 11.94134 | 12.844 | 0.522 | 6327 | <1.75 |
| 5219 | 132.70275 | 12.00243 | 12.797 | 0.553 | 6205 | 2.21 |
| 5675 | 132.83399 | 11.77831 | 12.755 | 0.559 | 6181 | 1.98 |
| 5562 | 132.81163 | 11.79006 | 12.805 | 0.572 | 6131 | 2.30 |
| 5644 | 132.82737 | 11.82270 | 12.647 | 0.608 | 5994 | <1.55 |
| 5573 | 132.81406 | 11.83739 | 12.823 | 0.565 | 6158 | 2.21 |
| 5825 | 132.86439 | 11.89076 | 12.756 | 0.592 | 6054 | 2.00 |
| 6076 | 132.92490 | 11.72707 | 12.69 | 0.564 | 6162 | 1.75 |
| 6107 | 132.93353 | 11.77351 | 12.75 | 0.758 | 5470 | <1.01 |
| 5993 | 132.90023 | 11.77605 | 12.722 | 0.683 | 5720 | 2.47 |
| 6177 | 132.95829 | 11.82540 | 12.647 | 0.581 | 6096 | <1.49 |
| 6077 | 132.92196 | 11.90817 | 12.76 | 0.580 | 6100 | 1.90 |
| 5996 | 132.89769 | 11.96578 | 12.826 | 0.775 | 5423 | <1.10 |
| 6408 | 133.04736 | 11.76044 | 12.889 | 0.817 | 5313 | <0.81 |
| 6224 | 132.97245 | 11.80585 | 12.705 | 0.575 | 6120 | 1.51 |
| 6313 | 132.99856 | 11.88273 | 12.79 | 0.584 | 6085 | 2.49 |
| 5951 | 132.88859 | 11.81431 | 12.767 | 0.577 | 6112 | <1.39 |

All listed stars have cluster membership probabilities greater than 0.5.
[a]ID, V, and B−V values are sourced from ref. 63. The original RA and DEC, initially in B1950, have been converted to J2000 using the astropy package[64].
[b]The original $T_{eff}$ and A(Li) values have been recalibrated to our $T_{eff}$ scale, enabling a direct comparison with our dataset. Only stars with M67 cluster membership >0.5 from ref. 65 are shown.
(This table is also provided as an Excel file and serves as the source data for Figs. 2 and 3).

plausible that metal-poor dwarfs are depleted by only about a factor of 3, due to their lower mass and potentially lower initial angular momentum. Existing models (e.g., [35,36]) predict that lower-mass and metal-poor stars experience less Li depletion, likely due to less efficient rotational mixing, compared to solar-type stars or those in NGC 188, where mixing is more pronounced.

## Method
### Observations
Data for candidate members of NGC 188, ranging from dwarfs with less than 1 solar mass up through the red giant branch, were obtained using the multi-fiber Hydra spectrograph mounted on the WIYN 3.5-m telescope, by using multiple configurations during different observing runs in 1995 November, 1996 April, 1997 June, 2001 February and March, 2002 April, and 2017 December. The echelle 316 l/mm grating was used, which provided a resolution of approximately $R = 13,000$ with the blue fibers and $R = 17,000$ with the red fibers, covering a range of approximately 400 Å centered near 6650 Å. In March 2001 and April 2002, we used the 31.6 l/mm KPNO coude grating, covering a wavelength range of 6700–6730 Å, to take

advantage of its higher throughput efficiency and slightly higher resolution ($R = 19,000$). This grating works well for studying the Li I feature at 6707.78 Å, but has an insufficient number of Fe I lines for the determination of metallicity; therefore, metallicity has been determined using data from the 316 l/mm grating[28].

### Sample selection
To focus on the topic of this study, we have carefully selected 96 turnoff and subgiant members from NGC 188, including only turnoff stars more evolved than $V = 15.4$ mag, and subgiants as bright as $V = 14.16$ mag. Assuming a distance modulus (m-M) = 11.48 mag and $E(B−V) = 0.09$ mag[28], and applying the extinction relation $A_V = 3.1 E(B−V)$[37], we transform the apparent V to $M_V$, and show the selected stars in Fig. 3. For comparison, we also show members of M67[23], transforming them assuming (m-M) = 9.75 mag and $E(B−V) = 0.04$ mag[29]. A study of subgiants and giants brighter than $V = 14.16$ mag and the apparent production of Li in some of these stars was presented in ref. 28, and main-sequence stars fainter than $V = 15.4$ mag will be discussed in future work.

### Data reductions and membership/multiplicity
Following similar procedures as described in ref. 28, we removed the instrumental signature from the spectra, calculated radial velocities for each night (provided in Supplementary Data 2), and used the combined spectra along with Gaia DR2[38] proper motion and parallax data, to assess multiplicity and membership of each individual star. In brief, cluster membership was determined using a combination of proper motions, parallaxes, and radial velocities. If a star's parallax and proper motions in RA and Dec lie within 2σ of the cluster's mean values, and its radial velocity is also within 2σ of the cluster average, we classify it as a member; otherwise, we consider it a non-member or unknown member. Multiplicity was determined based on radial velocity variations across different nights. We also cross-matched with ref. 39, who monitored the radial velocities of NGC 188 stars over many years, and found our classifications of both multiplicity and membership to be consistent with theirs. Using similar precepts, we designated 76 stars as SM ("single member"), 17 as BM ("binary member"), 1 as SM? ("single likely member"), and 2 as BM? ("binary likely member"). The proper motions and parallaxes from Gaia DR2 and DR3[40] are consistent and yield the same results. The fundamental stellar parameters are included in Supplementary Data 1, and the radial velocities for individual nights are included in Supplementary Data 2.

### Stellar atmospheres
The procedures follow closely those described in ref. 28. We have adopted their [Fe/H] of +0.064 ± 0.018 dex, and used their choice of a 6.3 Gyr Y²[41] isochrone with $E(B−V) = 0.09$ mag. We determined $T_{eff}$ using UBVRI photometry from refs. 42–44, cross-calibrated to a common photometric system. For each star, we computed all ten possible color indices from UBVRI and converted them to equivalent B−V values using empirical polynomials calibrated on cluster members. The final adopted B−V is the average of these ten B−V values, reducing random photometric errors to σ(B-V) about 0.01–0.02 mag. $T_{eff}$ was then derived by applying empirical color-$T_{eff}$ relations appropriate for each star's evolutionary stage. Given that our sample comprises both turn-off stars and subgiants, we tailored our approach by applying the color-$T_{eff}$ and microturbulence relations of ref. 28 for subgiants and giants with $T_{eff} < 5535$ K (($B−V)_0 > 0.706$ mag), and those of ref. 45 for subgiants and dwarfs with $T_{eff} > 5535$ K (($B−V)_0 < 0.706$ mag). The $T_{eff}$ uncertainties were propagated from the σ(B−V) errors. We determined surface gravity (log g) by comparing stellar positions to the Y² isochrone[46] that best matched our derived $T_{eff}$. The log g uncertainties were propagated from the $T_{eff}$ errors, while microturbulence uncertainties were derived from both $T_{eff}$ and log g errors.

## Li abundances

The procedures closely follow those described in refs. [28], [45]. Briefly, for each of the 96 stars, we create synthetic spectra of the Li I 6707.8 Å line using MOOG[47] to derive lithium abundances, $A(Li) = 12 + \log(N_{Li}/N_H)$. We distinguish between Li detections from $3\sigma$ upper limits guided by the precepts in ref. [48]: A(Li) for detections are determined through precise spectral synthesis of the Li line, while $3\sigma$ upper limits are calculated using local S/N ratios and line widths. We employed an optimized line list calibrated using high-resolution spectra of metal-rich giants (including $\mu$ Leo), with particular attention to weak blending lines near the Li doublet. We neglected $^6Li$ contributions given its expected over-depletion during pre-main-sequence and main-sequence evolution. The final A(Li) detections and upper limits are shown in Fig. 2, Table 1, and Supplementary Data 1. The A(Li) uncertainties combine errors propagated from stellar atmosphere parameters with the $1\sigma$ equivalent width measurement errors, added in quadrature. For stars with $3\sigma$ upper limits on A(Li), we do not report uncertainties as the upper limits themselves contain the measurement uncertainties.

## A(Li)-$T_{eff}$ slope in M67

The A(Li) data for M67 are shown in Table 1. In Fig. 2, we present the Li-$T_{eff}$ trend for M67 using both a linear fit to detections and a probabilistic model that incorporates upper limits. The solid green line shows a linear fit to the M67 detections alone, which yields a steep slope, partly due to the narrow $T_{eff}$ range over which Li is detectable (>6000 K). This detection-only fit may not capture the full behavior of the Li-$T_{eff}$ relation in M67, particularly at cooler temperatures where only upper limits are available. To illustrate the slope difference, we overlay a dashed green reference line adopting the same slope as the best-fit line for NGC 188. Although this line appears to follow both the M67 detections and upper limits, the latter provide only one-sided constraints. The true A(Li) values could lie just below the upper limits or be several orders of magnitude lower, making the line effectively a lower bound on the slope. This introduces significant uncertainty, particularly given the small number of upper limits and the complete absence of detections at cooler $T_{eff}$ (<6000 K).

While the true slope could in principle be as steep as the detection-only fit, the presence of upper limits at cooler $T_{eff}$ allows for the possibility that it might be shallower, though still steeper than the trend observed in NGC 188. We apply a hierarchical Bayesian linear regression using the linmix Python package to explore the probable slopes for M67, incorporating both detections and upper limits. The median slope from this analysis is shown as a green dash-dotted line in Fig. 2. The Li depletion trend in NGC 188 remains shallower than in M67 regardless of whether upper limits are included in the fit. However, the precise slope for M67 across a broader $T_{eff}$ range remains uncertain due to the current detection limit.

## MESA simulations

Our stellar evolution models are computed using the MESA, version 24.08.1[49,50]. All models start with the same initial A(Li) of 3.25 dex at the early pre-MS, based on the meteoritic value[30], and evolve through the entire pre-MS and MS, reaching the subgiant branch and continuing up to the base of the red giant branch.

Standard model: For NGC 188, we adopt a metallicity of [Fe/H] = +0.064 dex[28] and use the default MESA metal fraction[51] (`initial_zfracs = 3`). We use the `pp_and_cno_extras.net` nuclear network to include $^7Li$. Convective boundaries are set by the Ledoux criterion, and thermohaline mixing is included with $\alpha_{th} = 1$[52]. We adopt mixing-length theory[53] with $\alpha_{MLT} = 2$, and model semiconvection with $\alpha_{SC} = 0.1$[54]. Thermohaline mixing, although traditionally considered a non-standard process, is now commonly included in stellar evolution models and is adopted here for consistency. We test its impact by turning it on and off, and find that it has no visible effect on any of the

model lines. All four models use the same treatment for convection and mixing as described above, including thermohaline mixing, to ensure a consistent comparison.

Mass-loss model: we assume a constant mass-loss rate of $10^{-10} M_\odot$ yr$^{-1}$ [23].

Diffusion model: elemental diffusion is treated by solving the full Burgers equations (`diffusion_use_cgs_solver = .true.`) using diffusion coefficients from ref. [55].

Rotational mixing model: rotational mixing[56,57] incorporates dynamical shear[58], secular shear instability[59], Eddington-Sweet circulation[60], and the Goldreich-Schubert-Fricke instability. We set the ratio of turbulent viscosity to chemical diffusivity to $f_c = 1/30$[61], which reproduces the observed solar $^7Li$ abundance[9]. A zero-age main sequence rotational velocity of 29 km s$^{-1}$ provides the best match to NGC 188.

## Data availability

All data analyzed and generated in this study, including the reduced spectra obtained with the WIYN telescope and the MESA data, are publicly available[62]. The reduced spectra serve as the original data for deriving stellar atmospheric parameters and lithium abundances. The MESA data are included in the Fig. 2 Source Data file and can be reproduced following the procedures described in the main text. Additional data supporting the findings are provided within the article, its Supplementary Data files, and the Source Data files. The datasets generated during and/or analyzed during the current study are available from the corresponding author upon request. Source data are provided with this paper.

## Code availability

The MOOG stellar line analysis program can be accessed from the website: https://www.as.utexas.edu/~chris/moog.html. Stellar model atmospheres are obtained from the following source: http://kurucz.harvard.edu/grids.html. The MESA (Modules for Experiments in Stellar Astrophysics) stellar evolution code is available at: https://docs.mesastar.org/en/latest/.

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

## Acknowledgements

Q.S. is supported by the National Key R&D Program of China No. 2024YFA1611801, the Science and Technology Commission of Shanghai Municipality under Grant No. 25ZR1402244, and the Startup Fund for Young Faculty at Shanghai Jiao Tong University. C.P.D. is supported by the National Science Foundation (NSF) through grant AST-1909456. The authors thank the WIYN 3.5-m staff for helping us obtain excellent spectra. The authors also thank Dichang Chen and Xing Wei for helpful discussions on MESA models. This work has made use of data from the European Space Agency (ESA) mission Gaia (https://www.cosmos.esa.int/gaia), processed by the Gaia Data Processing and Analysis Consortium (DPAC, https://www.cosmos.esa.int/web/gaia/dpac/consortium). Funding for the DPAC has been provided by national institutions, in particular, the institutions participating in the Gaia Multilateral Agreement.

## Author contributions

Q.S. and C.P.D. designed the paper, conducted observations, analyzed the spectra, and prepared the paper. B.A.T. and B.J.A.T. helped with the scientific interpretation of the data. X.T.X. contributed to the development of the MESA model. A.S and J.R.K. also conducted observations. All authors thoroughly reviewed, provided feedback on, and reached a consensus on the manuscript.

## Competing interests

The authors declare no competing interests.
