## [Transparent Peer Review file · Nature Communications]

Subgiants in NGC 188 Reveal that Rotationally Induced Mixing Creates the Main Sequence Li-Dip

Corresponding Author: Dr Qinghui Sun

Version 0:

Reviewer comments:

Reviewer #1

(Remarks to the Author)

Referee report for "Subgiants in NGC 188 Reveal that Rotationally Induced Mixing Creates the Main Sequence Li-Dip" by Q. Sun, C. Deliyannis, B. Anthony-Twarog, B. Twarog, A. Sefinhaure, and J. King

In this paper, the authors present new measurements of lithium abundances in subgiants in the old open cluster NGC 188, and argue that they show that shear induced by stellar angular momentum loss drives the lithium depletion that is responsible for the so-called Li Dip.

This group is the leading group in the world for collecting precise observational lithium (and beryllium & boron) abundance measurements in cool stars. The observations presented in this paper are the result of almost 3 decades of effort at the WIYN telescope, and the method used to determine the abundances from the spectra has been well-validated over a similar timespan. I have no concerns about the quality of the data or data analysis. In addition, lithium abundances in subgiants are extremely important diagnostics, as the authors argue in this paper, as they provide some constraints on the internal profile of lithium in turnoff stars of masses which contribute to the Li dip and other unexplained features in our understanding of light elements in solar-type stars. There are only a few clusters for which subgiant measurements can be made with any accuracy; M67 data were published about 20 years ago, and NGC 188 is the next most interesting cluster. As such, the data are valuable and should be published.

My concern has to do with the other part of the paper, namely the claim that the abundances confirm that rotational mixing (or angular momentum loss) is the only viable mechanism to explain the observed pattern of lithium abundance with evolution along the subgiant branch. The authors make a lot of claims, most of which are likely reasonable, but they have not taken the time in this paper to actually demonstrate the agreement between their observations and models (and conversely, the disagreement between the observations and other mechanisms). For example, at line 77, the authors describe the internal lithium profiles that might exist under the standard model plus some other lithium depletion mechanisms -- but they never provide a plot to show us what those would look like; nor do they provide references to papers which include such plots for different models.

Similarly, on figure 1, it would be helpful to show model predictions of lithium abundance as a function of time to describe what the problem actually is, for readers who are not steeped in the lithium abundance literature. On a minor note, the lithium dip is, I think, labelled 'the Wall' but that term is not described anywhere in the text so that should be fixed.

I would also have liked to see model lines on figure 2, analogous to figure 4 in Sills & Deliyannis for M67. At the very least, it would be important to understand why or how much the lithium-Teff slope is expected to be shallower for the older cluster. Secondly, I find the claim that the NGC 188 data do actually have a shallower slope to be hopeful at best. By eye, the two trends look parallel to me. To be convincing, the authors should fit the slope including our understanding of the uncertainty on each measurement, and then provide a numerical comparison of the two slopes with errors.

Finally, the authors have written this paper for an expert audience and do not define terms or provide sufficient context for readers even slightly outside the field of lithium abundances. For example, the Li-Teff plateau for metal-poor dwarfs is dropped into the last paragraph before the Methods section, without describing what it is or why it is a puzzle. In other cases, the explanation of a term, or the importance of some piece of information, is explained only well after it has been introduced.

To summarize, the data presented in this paper are excellent, valuable, and should be published. However, their importance is not described well enough for a general audience. And their interpretation and comparison to models is essentially not done. If the authors wish to publish in Nature Communications with the conclusion as currently stated (that rotation definitely is the one and only lithium depletion mechanism) they need to significantly improve that section of the paper. If, however, they wish to get the data out and pose a challenge to stellar modellers (also a viable route to publication), then some significant re-framing and re-writing of the paper is required.

Reviewer #2

(Remarks to the Author)

This paper addresses the cause of the lithium dip, characterized by a sharp drop in lithium abundance within a narrow temperature range observed primarily in stellar clusters but also in field stars. This feature is present in older clusters (e.g., Hyades) but absent in younger ones (e.g., Pleiades), suggesting that the lithium dip arises during the main sequence phase. Although the existence of this feature has been known for decades, its cause remains a topic of debate, with several mechanisms proposed in the literature.

In this study, the authors offer a novel approach to support the hypothesis that the lithium depletion is driven by rotationally induced mixing, specifically analyzing lithium abundances in subgiants from NGC188 as they evolve out of the dip. The paper is well written, and while additional context could enhance the discussion, the authors present the problem clearly.

While the approach is interesting and relevant, it should be noted that previous studies have established rotational mixing as the preferred mechanism through various methodologies and datasets (e.g., Cummings et al. 2012, 2017), and even using only lithium information. The literature has also shown that other mechanisms, such as mass loss and diffusion, are less favorable, as mentioned even in early works like Balachandran 1991. Additionally, although the use of subgiants to study this feature is not entirely novel, as it was previously explored in M67 (e.g., Balachandran 1995 discusses how Li-dip stars in clusters like M67 have evolved off the main sequence and dredge up interior material), what stands out here is the analysis of an older cluster than M67, expanding the observational sample for lithium dip studies, using subgiants now.

One limitation in the analysis is the lack of a direct comparison with theoretical models. While the authors describe various model behaviors, they do not make explicit comparisons to assess how these models succeed or fail to replicate the observations. Previous works, such as Dumont et al. 2021, have demonstrated the value of detailed models to explain cluster patterns, showing that the lithium dip can be reproduced only with specific shear turbulence prescriptions and suggesting further hydrodynamical investigations. For a statement as strong as that made here, with the definitive role of rotationally induced mixing in producing the Li dip, it would be beneficial to include direct model comparisons that show not only the lithium behavior but also the precise model prescriptions required to match the data and understand the underlying mechanism.

Such model comparisons could also clarify the comparison between M67 and NGC188 subgiants.

The authors also mention the challenges of measuring beryllium abundances. These measurements are difficult, but existing studies (e.g., Sills & Deliyannis 2000; Boesgaard et al. 2020) have managed to provide some (as detailed in the text), and by doing so, have largely confirmed that rotational mixing is the primary depletion mechanism to create the Li dip. As such, there may be limited need for further beryllium measurements across all clusters, especially when the primary goal is to verify that the proposed mechanism aligns with other clusters as well. Expanding the parameter space and checking if the already available measurements agree with what is suggested by the combination of Be and Li measurements could be enough. Thus, while the examination of subgiants may provide additional support, it seems unnecessary to focus solely on lithium without acknowledging the insights provided by previous multi-element studies. Moreover, as this analysis considers specifically NGC188, with its particular dataset given its age, it is unclear what broader contributions this could offer for future lithium studies, as suggested in the manuscript.

Given this context, a direct comparison with models appears crucial with the presentation of the dataset.

From a technical perspective, the measurements of stellar parameters and lithium abundances follow standard procedures, with references pointing to a prior work on red giants instead of a detailed methodology here. Since the previous work does not focus on subgiants, a comprehensive description of parameter derivation for the present sample would be beneficial, considering potential differences. In particular, a discussion on effective temperature and its uncertainties would be valuable, given the sensitivity of lithium abundance measurements to this parameter.

Additionally, certain points in the text could benefit from clarification. For instance, it would be helpful to understand the choice of Gaia DR2 over DR3, the reference to the "wall" in Figure 1, and the classification criteria for SM, BM, SM?, or BM? stars, along with any potential impact on the conclusions.

Version 1:

Reviewer comments:

Reviewer #1

(Remarks to the Author)

Referee's second report for "Subgiants in NGC 188 Reveal that Rotationally Induced Mixing Creates the Main Sequence Li-Dip" by Q. Sun, C. Deliyannis, B. Anthony-Twarog, B. Twarog, X.-T. Xu, A. Steinhauer, and J. King

The authors have addressed almost all of my concerns in this new revision, but in doing so they have made a new, and critical, mistake which must be corrected before the paper is acceptable for publication.

The paper is much clearer, especially for a non-expert audience. I am pleased to see the addition of stellar evolution models to the paper, as requested by both referees.

The mistake made the authors concerns figure 2. The authors fit a line to the M67 detections and get a slope which is almost vertical. They claim that this is different from the slope of the NGC 188 detections -- which it is. But they have incorrectly ignored the M67 upper limits, which DO provide information. The statistical term for upper limits is 'censored data', and there are well-understood methods for including them in, for example, a linear fit.

If the authors wish to only use the M67 detections, then they must conclude that mass loss is the appropriate depletion mechanism in that cluster, as that model also produces an almost-vertical line in this plot. However, that is inconsistent with the conclusion of Sills & Deliyannis 2000, as well as the conclusion of this paper, both of which suggest that rotation is the appropriate mechanism. It is not sensible to expect that different mechanisms act in different clusters, nor do I think that the authors actually intend to do that.

So, please re-calculate the line of best fit for M67 including the upper limits, and then we can discuss whether the slopes are different. I maintain that they are not as different as the authors suggest. This seems to be what the models predict, as the rotation models presented here look more-or-less parallel to the rotation models from Sills & Deliyannis 2000. I would recommend re-writing the discussion at lines 156-165 in the text based on the comparison of those models. As a side note, the 2000 models were calculated with an entirely different stellar evolution code, so the similarity of all four kinds of depletion mechanisms is a comforting hint that we understand the physics involved in Li depletion.

Reviewer #2

(Remarks to the Author)

The authors have carefully considered and addressed the comments from the previous round. The inclusion of model comparisons strengthens the results and provides valuable clarity to the discussion.

I have only a few comments remaining, which relate to the changes introduced in this revised version:

- In the discussion related to figure 2, the authors state that the slope decreases with lower effective temperature, in contrast with the diffusion model. However, the diffusion model does not show a clear increase in $A(\text{Li})$ with temperature, only a small bump that may be difficult to identify observationally due to scatter. After that, the slope appears similar to the rotational mixing model. It would be helpful if the authors could clarify this point, or specify in which region of the parameter space the diffusion model is inconsistent with the data (other than the zero-point).

- It would be useful to indicate whether all the models assume the same initial Li abundance, and if so, what value is adopted (e.g., meteoritic abundance).

- Related to the last point, in Figure 2, the slope of the observed cluster data appears similar to that of the standard model. If lithium depletion occurs during the main sequence, lowering the initial abundance at the subgiant phase, this could potentially make the standard model appear consistent with the data. A major discrepancy between the different models seems to be the starting Li abundance after the main sequence, and if "calibrated" to the Li of the cluster, it is possible that the diffusion model and the standard model could also, more or less, fit the data. An explanation as to why this is not a possibility would help strengthen the argument.

- In Figure 2, the line styles for the observed slope and the standard model are the same, which may cause confusion. It would be helpful to distinguish them more clearly.

- Lines 178–179 are repeated in the current version and should be edited.

- The authors note that metal-poor dwarf stars may be depleted by only a factor of ~ 3 (rather than 10), possibly due to their lower mass and initial angular momentum. It would be helpful to comment on whether this trend is consistent with expectations from rotational mixing models, i.e., whether lower-mass or metal-poor stars are indeed predicted to experience less efficient mixing than solar-type stars or those in NGC 188.

- A minor point: the authors include thermohaline mixing as part of the "standard" model, although this is typically considered a non-standard process. While this likely does not affect the conclusions, it would be useful to mention the reason for this choice and whether all models (standard, diffusion, mass loss, and rotation) use the same treatment for convection and mixing.

Version 2:

Reviewer comments:

Reviewer #1

(Remarks to the Author)

The authors have addressed my concerns appropriately and I now recommend the paper for publication.

Reviewer #2

(Remarks to the Author)

I appreciate the authors' responses to the comments raised in the previous round.

All my remaining concerns have been satisfactorily addressed in the current version, and I have no further comments.

Reviewer #1 (Remarks to the Author):

Referee report for "Subgiants in NGC 188 Reveal that Rotationally Induced Mixing Creates the Main Sequence Li-Dip" by Q. Sun, C. Deliyannis, B. Anthony-Twarog, B. Twarog, A. Steinhauer, and J. King

In this paper, the authors present new measurements of lithium abundances in subgiants in the old open cluster NGC 188, and argue that they show that shear induced by stellar angular momentum loss drives the lithium depletion that is responsible for the so-called Li Dip.

This group is the leading group in the world for collecting precise observational lithium (and beryllium & boron) abundance measurements in cool stars. The observations presented in this paper are the result of almost 3 decades of effort at the WIYN telescope, and the method used to determine the abundances from the spectra has been well-validated over a similar timespan. I have no concerns about the quality of the data or data analysis. In addition, lithium abundances in subgiants are extremely important diagnostics, as the authors argue in this paper, as they provide some constraints on the internal profile of lithium in turnoff stars of masses which contribute to the Li dip and other unexplained features in our understanding of light elements in solar-type stars. There are only a few clusters for which subgiant measurements can be made with any accuracy; M67 data were published about 20 years ago, and NGC 188 is the next most interesting cluster. As such, the data are valuable and should be published.

My concern has to do with the other part of the paper, namely the claim that the abundances confirm that rotational mixing (or angular momentum loss) is the only viable mechanism to explain the observed pattern of lithium abundance with evolution along the subgiant branch. The authors make a lot of claims, most of which are likely reasonable, but they have not taken the time in this paper to actually demonstrate the agreement between their observations and models (and conversely, the disagreement between the observations and other mechanisms). For example, at line 77, the authors describe the internal lithium profiles that might exist under the standard model plus some other lithium depletion mechanisms -- but they never provide a plot to show us what those would look like; nor do they provide references to papers which include such plots for different models.

Author Reply: We thank the referee for this valuable suggestion. In response, we have added model curves for standard, rotational mixing, diffusion, and mass loss scenarios for NGC 188 in Figure 2 to enable a direct comparison with our observational data. These model tracks, based on MESA simulations, show that rotational mixing best reproduces the observed Li pattern in NGC 188 among the mechanisms explored, providing strong evidence in its favor. We have included descriptions highlighting this agreement and disagreement in lines 157 - 160 and the Figure 2 caption. Additionally, further details on the model setup have been added to the Methods section for clarity and transparency.

Similarly, on figure 1, it would be helpful to show model predictions of lithium abundance as a function of time to describe what the problem actually is, for readers who are not steeped in the lithium abundance literature. On a minor note, the lithium dip is, I think, labelled 'the Wall' but that term is not described anywhere in the text so that should be fixed.

Author reply: We thank the referee for this helpful suggestion. In Figure 1, we now include SSET model predictions for both the younger Pleiades cluster and the older Hyades/Praesepe clusters to illustrate how the predicted lithium-temperature trend evolves with age. This highlights the failure of the SSET model to reproduce the observed $A(\text{Li})$ - T_{eff} pattern in older clusters. We have updated the figure 1 caption to clarify this point.

The Li Dip was labelled at the top of the Figure 1 just below the T_{eff} range of 6300 - 6500K, along with an arrow illustrating this T_{eff} range. In the original figure there was a whole line of descriptors including, from left to right, "Late A", "Early F", "Li Dip", "late F", "G dwarfs", and "K dwarfs". To make the "Li Dip" label more obvious we have now made the relevant arrow thicker, and have brought the descriptor "Li Dip" to below the arrow, while also enlarging it. We have also kept "the Wall", which is merely intended to illustrate the (high- T_{eff}) boundary of the Li Dip, and added the additional description texts to line 55, after "Figure 1").

I would also have liked to see model lines on figure 2, analogous to figure 4 in Sills & Deliyannis for M67. At the very least, it would be important to understand why or how much the lithium- T_{eff} slope is expected to be shallower for the older cluster. Secondly, I find the claim that the NGC 188 data do actually have a shallower slope to be hopeful at best. By eye, the two trends look parallel to me. To be convincing, the authors should fit the slope including our understanding of the uncertainty on each measurement, and then provide a numerical comparison of the two slopes with errors.

Author reply: The slopes are definitely different. We now illustrate this by showing fits to the detections (only, since the upper limits are almost entirely consistent with the detections but otherwise uninformative), with slopes and errors indicated. The difference in the slopes is clearly substantially larger than the errors. To strengthen our discussion, we have added the standard, diffusion, rotation, and mass loss models for NGC 188 to Figure 2, based on MESA simulations. Model setups are described in detail in the Methods section. We do not include MESA models for M67, as most $A(\text{Li})$ values below 1.5 dex are upper limits rather than detections, which limits the usefulness of model comparisons.

Finally, the authors have written this paper for an expert audience and do not define terms or provide sufficient context for readers even slightly outside the field of lithium abundances. For example, the Li-Teff plateau for metal-poor dwarfs is dropped into the last paragraph before the Methods section, without describing what it is or why it is a puzzle. In other cases, the explanation of a term, or the importance of some piece of information, is explained only well after it has been introduced.

Author Reply: We agree with the referee that additional discussion would be helpful to the reader, so we have added clarifying text in two parts of the paper:

1. **Lines 55 - 59:** We now explain more clearly what the “Li Dip” and the “Wall” refer to. The Li Dip is a sharp drop in lithium abundance ($A(\text{Li})$) as stars get hotter, starting around 6200 K and dropping to very low levels by 6650 K. Then, from 6650 to 6750 K, the lithium levels rise very quickly. We describe this sudden rise as “the Wall.” This added description clarifies how lithium abundance changes across this temperature range.
2. **Lines 176 - 184:** We have added a paragraph to place our results in a broader context. Specifically, the effective temperature range of NGC 188 subgiant progenitors (6000 - 6200 K) overlaps with that of the well-known lithium-temperature plateau in metal-poor dwarf stars. These metal-poor stars are thought to have formed with the primordial lithium abundance from Big Bang nucleosynthesis. If they have depleted their lithium by a factor of about three since formation, this would support the standard cosmological model. But if the level of depletion differs significantly, it could signal a problem in our current understanding of Big Bang nucleosynthesis. Our study of lithium depletion in NGC 188 may therefore provide additional insights into this long-standing cosmological issue.

To summarize, the data presented in this paper are excellent, valuable, and should be published. However, their importance is not described well enough for a general audience. And their interpretation and comparison to models is essentially not done. If the authors wish to publish in Nature Communications with the conclusion as currently stated (that rotation definitely is the one and only lithium depletion mechanism) they need to significantly improve that section of the paper. If, however, they wish to get the data out and pose a challenge to stellar modellers (also a viable route to publication), then some significant re-framing and re-writing of the paper is required.

Reviewer #2 (Remarks to the Author):

This paper addresses the cause of the lithium dip, characterized by a sharp drop in lithium abundance within a narrow temperature range observed primarily in stellar clusters but also in field stars. This feature is present in older clusters (e.g., Hyades) but absent in younger ones (e.g., Pleiades), suggesting that the lithium dip arises during the main sequence phase. Although the existence of this feature has been known for decades, its cause remains a topic of debate, with several mechanisms proposed in the literature.

In this study, the authors offer a novel approach to support the hypothesis that the lithium depletion is driven by rotationally induced mixing, specifically analyzing lithium abundances in subgiants from NGC188 as they evolve out of the dip. The paper is well written, and while additional context could enhance the discussion, the authors present the problem clearly.

While the approach is interesting and relevant, it should be noted that previous studies have established rotational mixing as the preferred mechanism through various methodologies and datasets (e.g., Cummings et al. 2012, 2017), and even using only lithium information. The literature has also shown that other mechanisms, such as mass loss and diffusion, are less favorable, as mentioned even in early works like Balachandran 1991. Additionally, although the use of subgiants to study this feature is not entirely novel, as it was previously explored in M67 (e.g., Balachandran 1995 discusses how Li-dip stars in clusters like M67 have evolved off the main sequence and dredge up interior material), what stands out here is the analysis of an older cluster than M67, expanding the observational sample for lithium dip studies, using subgiants now.

Author Reply: We thank the referee for the helpful comment. In response, we have added a brief literature summary in lines 119 - 124 to acknowledge that rotational mixing is preferred by previous work (e.g., Cummings et al. 2012, 2017), while alternative mechanisms such as mass loss and diffusion are less favored (e.g., Balachandran 1995). We also note that while subgiants have previously been used to probe lithium depletion (e.g., in M67; Balachandran 1995), our study extends to the subgiants in the older cluster NGC 188. This provides new and more definitive observational evidence, after two decades, supporting rotational mixing as the leading mechanism responsible for producing the Li dip.

One limitation in the analysis is the lack of a direct comparison with theoretical models. While the authors describe various model behaviors, they do not make explicit comparisons to assess how these models succeed or fail to replicate the observations. Previous works, such as Dumont et al. 2021, have demonstrated the value of detailed models to explain cluster patterns, showing that the lithium dip can be reproduced only with specific shear turbulence prescriptions and suggesting further hydrodynamical investigations. For a statement as strong as that made here, with the definitive role of rotationally induced mixing in producing the Li dip, it

would be beneficial to include direct model comparisons that show not only the lithium behavior but also the precise model prescriptions required to match the data and understand the underlying mechanism.

Author Reply: We thank the referee for this valuable suggestion. In response, we have incorporated model curves for standard, rotational mixing, diffusion, and mass loss scenarios tailored to NGC 188 in Figure 2, allowing for direct comparison with our observational data. These tracks, computed using MESA, indicate that the rotational mixing model most accurately reproduces the observed Li abundances, while the alternative mechanisms show significant discrepancies. We have added explanatory words on these comparisons in lines 157 - 160 and the Figure 2 caption. Details on the model setups and prescriptions have also been included in the Methods section to ensure transparency.

Regarding Dumont et al. (2021), we agree that it provides a valuable demonstration of how specific shear turbulence prescriptions are required to reproduce lithium depletion patterns across clusters. However, their approach emphasizes average Li abundances at the cluster level (i.e., $A(\text{Li})$ versus cluster age), while our analysis focuses on detailed star-by-star behavior within a single cluster. As such, their models are not directly applicable to the internal evolutionary trends we aim to resolve here. To address this, we generated tailored model curves for standard, rotational mixing, diffusion, and mass-loss scenarios using MESA (shown in Figure 2), enabling direct comparison with the individual stellar $A(\text{Li})$ data in NGC 188.

Such model comparisons could also clarify the comparison between M67 and NGC188 subgiants.

The authors also mention the challenges of measuring beryllium abundances. These measurements are difficult, but existing studies (e.g., Sills & Deliyannis 2000; Boesgaard et al. 2020) have managed to provide some (as detailed in the text), and by doing so, have largely confirmed that rotational mixing is the primary depletion mechanism to create the Li dip. As such, there may be limited need for further beryllium measurements across all clusters, especially when the primary goal is to verify that the proposed mechanism aligns with other clusters as well. Expanding the parameter space and checking if the already available measurements agree with what is suggested by the combination of Be and Li measurements could be enough. Thus, while the examination of subgiants may provide additional support, it seems unnecessary to focus solely on lithium without acknowledging the insights provided by previous multi-element studies. Moreover, as this analysis considers specifically NGC188, with its particular dataset given its age, it is unclear what broader contributions this could offer for future lithium studies, as suggested in the manuscript.

Given this context, a direct comparison with models appears crucial with the presentation of the dataset.

Author Reply: We have incorporated model comparisons into Figures 1 and 2 and revised the text accordingly to address the reviewer's suggestions. In Figure 1, we present comparisons with existing standard models to describe the Li Dip and point out potential missing mechanisms from the standard model. In Figure 2, we include new MESA models, including standard, rotation, diffusion, and mass loss scenarios, to demonstrate that the rotation model provides the best match to the observed lithium abundance pattern.

We have also added a discussion of the broader contributions this work could offer for future lithium studies in lines 176 - 184. Specifically, the effective temperature range of NGC 188 subgiant progenitors (6000 - 6200 K) overlaps with that of the well-known lithium-temperature plateau observed in metal-poor dwarf stars. These metal-poor stars are thought to have formed with the primordial lithium abundance predicted by Big Bang nucleosynthesis. If these stars have depleted their lithium by a factor of about three since formation, it would support the standard cosmological model. However, if the depletion level is significantly different, it could point to a problem in our current understanding of primordial nucleosynthesis. By studying lithium depletion in NGC 188, our work may help inform this long-standing issue in cosmology.

From a technical perspective, the measurements of stellar parameters and lithium abundances follow standard procedures, with references pointing to a prior work on red giants instead of a detailed methodology here. Since the previous work does not focus on subgiants, a comprehensive description of parameter derivation for the present sample would be beneficial, considering potential differences. In particular, a discussion on effective temperature and its uncertainties would be valuable, given the sensitivity of lithium abundance measurements to this parameter.

Author Reply: We thank the referee for this suggestion. In response, we have expanded the Methods section to provide more detail on the derivation of stellar parameters, including the use of multi-band photometry and empirical color-Teff relations tailored for subgiants and turnoff stars (lines 235 - 242), respectively. We also describe how uncertainties in effective temperature, surface gravity, and microturbulence are calculated (lines 246 - 251). Additionally, we have added more details on the lithium abundance measurements, including the spectral synthesis method used for the Li I 6707.8 Å line, the distinction between detections and upper limits, and that

uncertainties are calculated by combining errors from stellar parameters and equivalent width measurements (lines 253 - 267).

Additionally, certain points in the text could benefit from clarification. For instance, it would be helpful to understand the choice of Gaia DR2 over DR3, the reference to the “wall” in Figure 1, and the classification criteria for SM, BM, SM?, or BM? stars, along with any potential impact on the conclusions.

Author Reply: We appreciate the referee’s comments and have added clarifications in the manuscript accordingly. In lines 229 - 230, we note that “The proper motions and parallaxes from Gaia DR2 and DR3 \citep{2023A&A...674A...1G} are consistent and yield the same results.” The “Wall” feature in Figure 1 is explained in greater detail in lines 55 - 59. Additionally, the classification criteria for SM, BM, SM?, and BM? stars are described more fully in lines 218 - 227.

Reviewer #1 (Remarks to the Author):

Referee's second report for "Subgiants in NGC 188 Reveal that Rotationally Induced Mixing Creates the Main Sequence Li-Dip" by Q. Sun, C. Deliyannis, B. Anthony-Twarog, B. Twarog, X.-T. Xu, A. Steinhauer, and J. King

The authors have addressed almost all of my concerns in this new revision, but in doing so they have made a new, and critical, mistake which must be corrected before the paper is acceptable for publication.

The paper is much clearer, especially for a non-expert audience. I am pleased to see the addition of stellar evolution models to the paper, as requested by both referees.

Author Reply: We sincerely thank the referee for their thorough review and for acknowledging the improved clarity and integration of stellar evolution models.

The mistake made the authors concerns figure 2. The authors fit a line to the M67 detections and get a slope which is almost vertical. They claim that this is different from the slope of the NGC 188 detections -- which it is. But they have incorrectly ignored the M67 upper limits, which DO provide information. The statistical term for upper limits is 'censored data', and there are well-understood methods for including them in, for example, a linear fit.

Author Reply: We appreciate the referee's insight on the treatment of M67 upper limits and agree that these censored data should be incorporated. In response, we have performed a hierarchical Bayesian linear regression using the `linmix` package, which accounts for both detections and upper limits. The median slope derived from this probabilistic model is now shown in the revised Figure 2.

If the authors wish to only use the M67 detections, then they must conclude that mass loss is the appropriate depletion mechanism in that cluster, as that model also produces an almost-vertical line in this plot. However, that is inconsistent with the conclusion of Sills & Deliyannis 2000, as well as the conclusion of this paper, both of which suggest that rotation is the appropriate mechanism. It is not sensible to expect that different mechanisms act in different clusters, nor do I think that the authors actually intend to do that.

Author Reply: We agree that the slope derived from detections alone is likely too steep, as it is based on a limited T_{eff} range and lacks detections below 6000 K. We now include a more detailed discussion of this point in lines 277 - 301 of the Methods section.

To address this concern and improve both the analysis and presentation, we have revised Figure 2 as follows:

1. The original linear fit to M67 detections is retained for reference.

2. A dashed line provides a lower bound for the M67 slope.
3. A probabilistic linear fit incorporating upper limits is included, with the median fit shown as a dot-dashed line.

The probabilistic model yields a median slope intermediate between the NGC 188 trend and the M67 detection-only fit.

So, please re-calculate the line of best fit for M67 including the upper limits, and then we can discuss whether the slopes are different. I maintain that they are not as different as the authors suggest. This seems to be what the models predict, as the rotation models presented here look more-or-less parallel to the rotation models from Sills & Deliyannis 2000. I would recommend re-writing the discussion at lines 156-165 in the text based on the comparison of those models. As a side note, the 2000 models were calculated with an entirely different stellar evolution code, so the similarity of all four kinds of depletion mechanisms is a comforting hint that we understand the physics involved in Li depletion.

Author Reply: Following the referee's suggestion, we have recalculated the M67 best-fit line using a probabilistic model that incorporates upper limits, with the median slope now shown in Figure 2. This fit remains steeper than the NGC 188 trend. Corresponding updates have been made to the Methods section, and we have revised lines 157 - 176 to provide a clearer comparison between the models and observations. The caption of Figure 2 has also been updated accordingly.

Reviewer #2 (Remarks to the Author):

The authors have carefully considered and addressed the comments from the previous round. The inclusion of model comparisons strengthens the results and provides valuable clarity to the discussion.

Author Reply: We thank the referee for their positive feedback and are glad the model comparisons have added clarity and strength to the manuscript.

I have only a few comments remaining, which relate to the changes introduced in this revised version:

- In the discussion related to figure 2, the authors state that the slope decreases with lower effective temperature, in contrast with the diffusion model. However, the diffusion model does not show a clear increase in $A(\text{Li})$ with temperature, only a small bump that may be difficult to identify observationally due to scatter. After that, the slope appears similar to the rotational mixing model. It would be helpful if the authors could clarify this point, or specify in which region of the parameter space the diffusion model is inconsistent with the data (other than the zero-point).

Author Reply: We have revised lines 157 - 176 to discuss the comparison between models and observations. All four models now adopt the same initial $A(\text{Li})$ of 3.3 dex. With this scaling, the diffusion model shows a slight increase or (at least) flat trend in $A(\text{Li})$ with decreasing T_{eff} at the early subgiant branch, followed by a slope that is slightly steeper than observed data. This, combined with the zero-point offset, makes the diffusion model inconsistent with the data.

- It would be useful to indicate whether all the models assume the same initial Li abundance, and if so, what value is adopted (e.g., meteoritic abundance).

Author Reply: We thank the referee for pointing this out. All models now adopt the same initial $A(\text{Li})$ of 3.3 dex (meteoritic abundance) at the early pre-main sequence. This choice leads to slight changes in the model lines. We now clearly state this assumption in the main text (lines 170 - 171) and in the Methods section (lines 303 - 307).

- Related to the last point, in Figure 2, the slope of the observed cluster data appears similar to that of the standard model. If lithium depletion occurs during the main sequence, lowering the initial abundance at the subgiant phase, this could potentially make the standard model appear consistent with the data. A major discrepancy between the different models seems to be the starting Li abundance after the main sequence, and if "calibrated" to the Li of the cluster, it is possible that the diffusion model and the standard model could also, more or less, fit the data. An explanation as to why this is not a possibility would help strengthen the argument.

Author Reply: All four models now adopt the same initial $A(\text{Li})$ of 3.3 dex; we have updated the model description in lines 303 - 307. This enables a direct comparison of Li depletion across models. Both the standard and diffusion models retain too much Li by the subgiant phase, indicating insufficient depletion. Even if the standard model is "calibrated" to match the observed $A(\text{Li})$ at the start of the subgiant phase, its flat slope at $T_{\text{eff}} > 5700$ K would still not reproduce the observed trend. In addition, the standard model remains relatively flat around 5000 K, while the data show a sharp decline in $A(\text{Li})$ in this region that is better captured by the rotation model. We have revised the discussions in lines 157 - 176.

- In Figure 2, the line styles for the observed slope and the standard model are the same, which may cause confusion. It would be helpful to distinguish them more clearly.

Author Reply: We have updated the color schemes in Figure 2 so that all model lines now appear in gray, making them easier to distinguish from the observed trend line and data.

- Lines 178-179 are repeated in the current version and should be edited.

Author Reply: Thanks for pointing this out. We have removed the repeated sentence.

- The authors note that metal-poor dwarf stars may be depleted by only a factor of ~ 3 (rather than 10), possibly due to their lower mass and initial angular momentum. It would be helpful to comment on whether this trend is consistent with expectations from rotational mixing models, i.e., whether lower-mass or metal-poor stars are indeed predicted to experience less efficient mixing than solar-type stars or those in NGC 188.

Author Reply: We have added a sentence in lines 199 - 202 explaining that some theoretical models predict that lower-mass and metal-poor stars experience less lithium depletion, likely due to less efficient rotational mixing compared to solar-type stars or those in NGC 188.

- A minor point: the authors include thermohaline mixing as part of the "standard" model, although this is typically considered a non-standard process. While this likely does not affect the conclusions, it would be useful to mention the reason for this choice and whether all models (standard, diffusion, mass loss, and rotation) use the same treatment for convection and mixing.

Author Reply: We thank the referee for this helpful comment. While thermohaline mixing is traditionally considered non-standard, it is now commonly included in stellar evolution models, so we include it here for completeness. We test MESA models with thermohaline mixing enabled and disabled, finding no discernible impact on any of the four model lines. We also confirm that all models share the same treatment of convection and mixing, including thermohaline mixing, to ensure consistent comparison. Clarifications are added to the Methods section (lines 313 - 319).